# From NSAIDs to Glucocorticoids and Beyond

**DOI:** 10.3390/cells10123524

**Published:** 2021-12-14

**Authors:** Ajantha Sinniah, Samia Yazid, Rod J. Flower

**Affiliations:** 1Department of Pharmacology, Faculty of Medicine, University of Malaya, Kuala Lumpur 50603, Malaysia; 2Trio Medicines Ltd., Hammersmith Medicines Research, London NW10 7EW, UK; samia_yazid@hotmail.com; 3Biochemical Pharmacology, William Harvey Research Institute, Queen Mary University of London, London EC1M 6BQ, UK; r.j.flower@qmul.ac.uk

**Keywords:** nonsteroidal anti-inflammatory drugs (NSAIDs), glucocorticoids, annexin A1, inflammation

## Abstract

Our interest in inflammation and its treatment stems from ancient times. Hippocrates used willow bark to treat inflammation, and many centuries later, salicylic acid and its derivative aspirin’s ability to inhibit cyclooxygenase enzymes was discovered. Glucocorticoids (GC) ushered in a new era of treatment for both chronic and acute inflammatory disease, but their potentially dangerous side effects led the pharmaceutical industry to seek other, safer, synthetic GC drugs. The discovery of the GC-inducible endogenous anti-inflammatory protein annexin A1 (AnxA1) and other endogenous proresolving mediators has opened a new era of anti-inflammatory therapy. This review aims to recapitulate the last four decades of research on NSAIDs, GCs, and AnxA1 and their anti-inflammatory effects.

## 1. Introduction

### Inflammation and Pain

Historically, descriptions of inflammation date back to the ancient Egyptian and Greek cultures, although these early perceptions were largely based upon intuition rather than organised scientific investigation. *Hippocrates* regarded inflammation as the early component of the healing process after tissue injury, whilst *Aulus Cornelius Celsus*, in the first century, is often credited with describing the main four cardinal signs of inflammation: redness (*rubor*), warmth (*calor*), swelling (*tumor*), and pain (*dolor*) [1]. A fifth sign of inflammation, loss of function (*functio laesa*), was introduced in the 17th century by the Roman physician *Galen* [2]. An interesting observation was recorded by Virchow in 1871, who identified a cellular component of the inflammatory response, paving the way for cellular pathology [3]. These early unmethodical observations provided the framework for subsequent critical investigation by scientists, leading to conceptual shifts in the mechanistic understanding of the context and role of inflammation.

When tissues are damaged or infected, inflammatory mediators are released, resulting in arteriolar dilation, causing the area to become red and hot. Furthermore, formation of gaps between venules and capillaries initiates the passage of fluid and cells. Cell permeability is increased due to the fluid leakage caused by oedema. Cellular influx, on the other hand, is due to the process of marginalisation, fixation, and rolling of leukocytes together with the endothelial cells that border the lesion site. Migration of cells begins, and subsequently, interaction of specifically expressed adhesion molecules would trigger the release of inflammatory mediators that induce pain [4,5]. Nociceptors can eventually become sensitised to the signals they receive, causing allodynia (pain from nonpainful stimuli) and hyperalgesia (exaggerated pain from a painful stimulus) [6,7].

Research into inflammation initiated by either physiological or pathological immune responses is constantly evolving [8]. The advent of new disciplines, such as epigenetics, cellular metabolism, aging and senescence, and neurological and immunological interactions, will undoubtedly offer new insights into pathways of inflammation, which will facilitate the generation of novel therapeutics [9].

One of the most significant shifts in our understanding of inflammation was the observation that the resolution of inflammation is not just a passive process: proinflammatory mediators changing to ‘proresolving’ mediators trigger a negative feedback loop, dampening inflammation through an active and highly regulated process. Serhan and colleagues identified endogenous lipid molecules that mediate the resolution of inflammation by inhibiting neutrophil recruitment and extravasation and increase the phagocytosis of apoptotic neutrophils, leading to the clearance of inflammation and promoting tissue repair [10,11]. The inflammatory response is terminated when noxious stimuli are removed; thereafter, a damaged tissue undergoes repair and dead cells or debris are cleared, and the resolution of inflammation is initiated by the release of ‘immunoresolvent’ molecules, such as the protein annexin A1 (AnxA1), and lipidic resolvins, protectins, lipoxins, and maresins [12].

Amongst the five cardinal signs of inflammation, pain is one of the main reasons why patients seek drug intervention [13]. The ancient Greek author *Aeschylus* famously quoted, ‘*Who, except the gods, can live time through forever without any pain?*’ At least 10% of the world’s population is affected by chronic pain conditions [14], and 1 in 10 patients will be diagnosed annually with debilitating chronic pain [15].

The key concept that pain is a disease can be traced to the first decade of the 20th century, when Sherrington defined pain sensation as ‘the physical adjunct of an imperative, protective reflex’ [16]. However, in 1953, in a paradigm shift begun by John Bonica, the founding father of pain management, workers began to consider ‘pain as a biologically protective tool that eludes its adaptive function and turns pathological’. In his book ‘*The Management of Pain’*, Bonica distinguished normal from abnormal pain, which, if it persists, becomes pathological and modifies the biological signalling pathway. This can have a negative impact on the quality of life, requiring patients to heavily rely upon therapeutic pain management [17].

Pain is a tangible, unpleasant sensation and a complex personal encounter that involves both physiological and emotional factors [18]. Acute pain signals the possibility of potential tissue damage and inflammation. However, this usually diminishes quickly during the resolution phase of inflammation [19]. Nonetheless, in some individuals, the acute pain progresses to become chronic pain. This is defined as pain that outlasts the inflammation or injury beyond 3 months. Common causative factors of chronic pain are old age, trauma, injuries, acquired immune deficiency syndrome, autoimmune disease, and cancer [20].

## 2. Anti-Inflammatory Pharmacology

Anti-inflammatory plants figure prominently in ancient herbals, but the advent of truly effective synthetic drugs really dates from the end of the 19th century. The vast majority of these drugs were discovered by mere chance, through empirical observation or by mistake. They constitute a list that makes for somewhat embarrassing reading.

Take gold, for example. This was introduced in the 1920s in the erroneous belief that rheumatoid arthritis (RA) was somehow related to tuberculosis [21]. Additionally, penicillamine was discovered in the 1950s in the urine of patients who had received penicillin and tested in arthritis in the 1960s on the basis that it could counteract a rheumatoid factor in vitro [22]—although, as it turned out, it did not do this when given orally.

The antimalarial action of chloroquine was discovered in the 1930s, and the drug was tested in arthritis 20 years later apparently because another antimalarial, mepacrine, had proved to be useful [23]. Another repurposed drug was methotrexate. This was originally developed in the 1950s as a superior folate antagonist for use in cancer chemotherapy. It was first tested in RA in the early 1970s on the basis that a previous folate antagonist, aminopterin, had been shown to be active in psoriasis two decades earlier [24]. Methotrexate is a drug that is now very much in vogue again as a supplement to treatment with biologics, but how does it work? No lab has done more to elucidate this problem than the Cronstein group at the NYU School of Medicine, and their findings that methotrexate can liberate adenosine, thereby indirectly activating inhibitory purinergic receptors, have given us another insight into the complexity of anti-inflammatory drug action [25,26].

### 2.1. NSAIDs

It is intriguing to recall that *Hippocrates*, a Greek physician who lived in 3500 B.C., had already described the beneficial actions of extracts of willow bark and leaves in fever and inflammation [27,28]. This was ‘rediscovered’ in the UK in the late 17th century, and the active ingredient of willow bark was subsequently identified as salicylic acid. Therefore, when Kolbe synthesised this compound in 1859 by carboxylating phenol, it was tested in RA and other inflammatory conditions, becoming a best seller in its day [29].

The discovery of aspirin was said to be prompted by Felix Hoffmann’s rheumatic father, who had urged his son to produce a medicine that lacks the unpleasant effects of sodium salicylate. Hoffmann, who was a chemist at Bayer, went on to acetylate the phenol group of salicylic acid and produced pure stable acetylsalicylic acid (ASA) [30]. This compound was named aspirin and was developed by Bayer in the late 1890s as a competitor for salicylate—giving it the dubious honour of becoming what was probably the first ‘me-too’ drug [29]. It was enormously successful and is still manufactured and consumed in prodigious quantities around the world. The mere availability of aspirin as a model anti-inflammatory drug with a very characteristic profile of biological effects led to the development of other drugs with similar actions: phenylbutazone was introduced in the 1940s, fenamates in the 1950s, indomethacin in the 1960s, propionates in the 1970s, and oxicams in the 1980s [31]. These nonsteroidal anti-inflammatory drugs (NSAIDs) are often known as ‘aspirin-like drugs’ because their pharmacology is broadly similar to that of aspirin, the ‘archetypal’ NSAID. NSAIDs are structurally distinct, although most are carboxylic acids.

However, the actual mechanism of action of these drugs remained unknown until the early 1970s, when John Vane and his colleagues discovered the mechanism of action of aspirin and other NSAIDs [32], thereby paving the way to the development of novel anti-inflammatory drugs [33]. Vane discovered that aspirin blocked the action of the cyclooxygenase enzyme (COX-1), a constitutive enzyme that induces the formation of prostaglandins [32]. Since these lipids had been implicated in producing pain, fever, and inflammation, as well as in preventing gastric damage, this mechanism provided a comprehensive explanation for the well-known analgesic, anti-inflammatory, antipyretic, and gastrotoxic actions of aspirin (although a direct toxic action of NSAIDs on the gut may be at least in part mediated through an ‘uncoupling’ effect on epithelial mitochondria).

After Vane’s breakthrough discovery in 1971, he wrote that he had the idea whilst reviewing an experiment in which the release of ‘rabbit aorta contracting substance’ (RCS; later shown to be a mixture of eicosanoids and leukotrienes) from guinea pig and dog lung was inhibited by aspirin. Considering that RCS might be an intermediate in prostaglandin synthesis, he wrote, *“… a logical corollary was that aspirin might well be blocking the synthesis of prostaglandins”*. Vane and colleagues demonstrated that aspirin, indomethacin, and sodium salicylate blocked the synthesis of prostaglandins in cell-free systems and in isolated perfused dog spleen ex vivo. A further study in human volunteers taking a therapeutic dose of aspirin showed that the prostaglandin levels were reduced in seminal fluid samples and aggregating platelets, confirming that NSAIDs were able to inhibit the synthesis of prostaglandins by directly targeting the COX enzyme, both in vitro and in vivo [34].

The next major breakthrough in the NSAID field occurred in the early 1990s, when an inducible isoform of COX, termed COX-2, was discovered [35]. This was mooted to be the most significant isoform of the COX enzyme in inflammation but less important as a gastroprotective enzyme in the stomach. The conventional NSAIDs of the day were found to be unselective inhibitors acting upon both COX-1 and COX-2 to various extents, perhaps explaining why they had mixed therapeutic and toxic effects. This led, in turn, to the development of a new concept: that selective inhibitors of COX-2 should have superior anti-inflammatory but less adverse gastrointestinal effects than the traditional (unselective) NSAIDs [36]. The subsequent discovery of ‘coxibs’, more selective inhibitors of COX-2, such as celecoxib, rofecoxib, lumiracoxib, and valdecoxib, provided further options for treating patients who suffered from gastric irritation, although, disappointingly, they did not eliminate the problem altogether. Both the original NSAIDs and the newer coxibs remain the mainstay of anti-inflammatory drug therapy for pain and swelling in osteoarthritis and RA; acute inflammatory conditions such as sports injuries, fractures, and soft tissue injuries; and postoperative, headache, migraine dental, menstrual pain, and some acute trauma conditions [37].

Fifty years since the Cox theory of aspirin action was established, the new millennium had brought about a re-evaluation of our understanding of the therapeutic and side effects of these drugs; the explanations of NSAID cardiotoxicity and gastrointestinal toxicity are still controversial topics. Although the exact mechanism of action on how these drugs target cancer cells is not well elucidated, studies have shown that aspirin prevents colorectal cancer through the inhibition of NF-κB and signalling pathways [38]. New targets of aspirin and salicylic acid have been proposed, such as cyclin A2 and CDK2, among others, as a preventive action mechanism against cancer [39]. Conversely, aspirin has been shown to acetylate the inflammatory COX-2 isoform, leading to the production of proresolving lipid mediators, derived from omega-3 [40]. Interestingly, aspirin had been associated with a slower decline in several cognitive domains, particularly in Alzheimer’s disease [41], by targeting the peroxisome proliferator gamma (PPAR-γ) nuclear transcription factor [42] and via the downregulation of COX-1 and COX-2 [43].

NSAID drug classification is usually based on chemical structure and COX selectivity; nonselective inhibitors include acetylated salicylates (aspirin), nonacetylated salicylates (diflunisal), acetic acids (indomethacin, diclofenac), propionic acids (ibuprofen, naproxen), enolic acids (piroxicam, meloxicam), and the more selective COX-2 inhibitors (celecoxib, etoricoxib) [31]. Many NSAIDs are available in a variety of formulations, such as tablets, injections, and gels, and several are available in pharmacies without prescription. A derivative of aspirin, nitric-oxide-donating aspirin (NO-ASA), modified by inserting covalently a NO-donating moiety to the traditional aspirin, has been shown to render enhanced efficacy, potency, and increased safety profile [44].

Even though Vane’s concept had led to a simple yet robust in vitro screening for putative anti-inflammatory compounds that inhibit a COX enzyme [45], several anomalies were noted. For instance, paracetamol (acetaminophen) is a very prominent drug that was introduced in the late 1800s as an antipyretic drug and also possesses analgesic effects, like other aspirin-like drugs [46,47]. However, unlike the latter, paracetamol has very little anti-inflammatory activity and is completely devoid of any gastric or platelet effects [48]. In accordance with its therapeutic profile, paracetamol was found to be more effective against crude brain COX preparations compared with those isolated from peripheral tissues, such as the spleen, suggesting a putative explanation for the selectivity of its therapeutic action and the notion that there were several forms of the enzyme that were formulated [47]. In experimental animals, analgesia and antipyresis with paracetamol are accompanied by the reduction in prostaglandin synthesis in the central nervous system [47,49].

As paracetamol is only a weak inhibitor of COX-1 and COX-2 activities, its pharmacological actions could not be explained by the inhibition of these enzymes [47]. Interestingly, COX-3, which was identified as a splice variant of COX-1 in 2002 in canine tissues, was shown to be inhibited by paracetamol. It was thus hoped that the discovery of COX-3 might provide a neat explanation for the pharmacological actions of paracetamol. However, the existence of COX-3 in rodent and human tissues is debated (due to the fact that retention of intron-1 results in an out-of-reading frame sequence), despite some evidence in the literature on its possible expression in these species along with the demonstration that a COX-1 variant is involved in the thermoregulation in normothermic animals [50]. Whilst being a weak anti-inflammatory drug, paracetamol produced the same effect as NSAIDs at the latter phase of acute inflammation, in that it prevented the resolution of the inflammatory reaction (our unpublished observations). This finding echoes a randomised Strategies for Prescribing Analgesics Comparative Effectiveness (SPACE) study that was carried out to compare chronic pain management in patients with hip arthrosis and knee and back pain. The patients were administered with NSAIDs + adjuvants (nonopioid) and opioid therapy for 12 months [51]. The results indicate that NSAIDs alleviate pain better than the opioids, which reflects on pharmacological chronic pain management. On the contrary, some preliminary studies have shown that NSAIDs may interfere with some endogenous proresolving lipid mediators, hence disrupting the wound healing process, specifically postsurgery [52,53,54].

### 2.2. Glucocorticoids

A century and half ago, Thomas Addison, whilst working at Guy’s Hospital in London, first described the adrenal insufficiency syndrome that now bears his name. He realised that this was a disease involving the ‘suprarenal capsules’, the adrenal glands, a discovery that earned him the accolade of the ‘father of endocrinology’ [55]. However, it was the later discovery of cortisol by Hench and his colleagues and their demonstration that this synthetic steroid has profound anti-inflammatory effects in patients with RA that revolutionised the field of rheumatology as well as many other aspects of medicine [56].

The American rheumatologist Philip Hench made an unusual observation in 1929. He noticed that pregnant patients or those suffering from jaundice experienced a dramatic relief from their arthritic symptoms. He postulated that an unknown substance (later termed ‘Compound E’) circulating in the blood during pregnancy or jaundice could account for the observed symptomatic relief [57,58].

In 1948, Hench administered a synthetic pharmaceutical formulation of Compound E (subsequently identified as cortisone) to patients with RA. This produced a ‘miraculous’ improvement, in which the patients experienced significant reduction in joint stiffness, decreased articular pain, and enhanced mobility of the joints [58,59,60]. Subsequent problems with overdosing led to a period during which cortisone fell out of favour, but today, GCs are one of the most widely prescribed drugs, with a net global market worth of more than USD 10 billion per year [61]. Therefore, revolutionary were the effects of cortisol that the treatment of inflammation is often humorously separated into ‘before cortisol’ (BC) and ‘after cortisol’ (AC) eras. Hench’s discovery had a tremendous impact, earning him a Nobel Prize in 1950.

Unfortunately, glucocorticoids (GC) entail a heavy burden of side effects, and if treatment is discontinued, the symptoms may return or even rebound [62]. Two major drawbacks attend GC treatment: First, long-term treatment can cause osteoporosis, diabetes, hypertension, skin hypertrophy, adrenal suppression, and glaucoma [63]. Second, GC resistance may develop, reducing the benefits of GC therapy [64]. However, newly synthesized GC drugs, such as methylprednisolone, betamethasone, and fluorinated dexamethasone, boast stronger anti-inflammatory and immunosuppressive potencies but weaker mineralocorticoid effects as compared with cortisone [65]. Of particular relevance to the current pandemic, dexamethasone has been highlighted as an effective medicine against severe coronavirus disease 2019 (COVID-19). Although the suppression of a damaging immune response by glucocorticoids could be deemed beneficial, it does not prevent viral replication and, in fact, impedes the ability of the host to fight infection. However, it has been reported that SARS-CoV-2 virus induces a coincidental glucocorticoid insensitivity in infected cells, possibly initiating the activation of the immune system and therefore modulating the intracellular actions of glucocortiocids, hence contributing to the therapeutic effects of dexamethasone on severe cases [66].

Targeted delivery of GCs directly to the site of inflammation can reduce the overall dose necessary to achieve a therapeutic effect, and optimal timing of GC administration can reduce the tendency to cause adrenal suppression. Our increased knowledge of the mechanistic action of GCs, especially the recognition of rapid nongenomic effects, has contributed to the improved GC benefit/risk ratio [67,68].

#### GC Receptor/Mechanism of Action

The mechanistic actions of GCs are largely mediated through the classic GC receptor (GR), which is constitutively expressed by all cell types, although there is a variable spectrum of GC sensitivity and biological responses, between tissue types. The two protein isoforms of GR that were cloned are hGRα and hGRβ [69]. The former predominantly resides in the cytoplasm in its inactive state, whereby it is sequestered by the chaperone complexes consisting of heat shock protein (hsp), immunophilins, and other factors that prevent its degradation, whilst the latter is located in the nucleus and acts as a natural dominant negative inhibitor of hGRα [70]. hGRα associates with hsp90 and hsp56 to maintain the receptor in a conformation that will bind to the GC with high affinity but will not bind to DNA [71]. Upon activation, GR exerts its acute anti-inflammatory effects via either genomic or nongenomic mechanism by activating the AnxA1 pathway and other cellular effects [72].

The genomic mechanism occurs upon ligand binding to hGRα, resulting in the dissociation of chaperone proteins, which translocate to the nucleus and bind to glucocorticoid response elements (GRE) to either transactivate anti-inflammatory genes such as IL-10, AnxA1, and MAPK or transrepress proinflammatory transcription factors such as AP-1 and NF-kB, resulting in the downregulation of cytokine production. This action is governed by phosphorylation signals on GR, which contains multiple phosphorylation sites in the N-terminal domain [73,74].

The dogma that the transactivation of a transcription factor of the homodimeric GR is the main culprit that contributes to the side effects of GCs, and that transrepression is the main contributing factor of GR’s anti-inflammatory properties, has been challenged [75]. The reason for this ambiguity is that there are numerous characterised genes, including AnxA1, IL-10, GC-induced leucine zipper (GILZ), IκBα, genes coding for MKP-1, secretory leukoprotease inhibitor (SLPI), and type II (decoy) IL-1, that are being upregulated by GR that do possess distinctive anti-inflammatory roles [76], which could amplify the anti-inflammatory actions of GCs.

Apart from genomic effects, GCs also exert acute nongenomic actions that occur immediately after GR ligation [77]. The transgenic murine model expressing mutant form of GR that is not capable of translocation to the nucleus upon ligand binding, was still viable, suggesting that there may be a substantial compensatory key effects that are mediated through non-genomic pathway such cytosolic-GR actions [78].

### 2.3. AnxA1 and Anti-Inflammation

NSAIDs work by inhibiting prostaglandin synthesis, and for a while, it seemed, therefore, that prostaglandins were key mediators of inflammation. However, what about other anti-inflammatories? Did they act in the same way? Apparently not; GCs and the other so-called disease-modifying drugs, such as gold and penicillamine, were inactive when tested by Flower et al. (1972) in an in vitro system for measuring prostaglandin synthesis [49]. At the time, this was a puzzle: if they did not work through the inhibition of prostaglandin generation, then what was their mechanism of action, and what did this say about the role of these lipids in inflammation? Another observation supplied an important clue: whilst the GCs were inactive when tested in cell-free COX assays, they were active as inhibitors of PG generation in vivo or when intact cells were present. Therefore, what was the answer to this conundrum?

Our laboratory pioneered the idea that GCs acted by stimulating the synthesis and release of a second messenger of GC action, which duplicated many of their effects. This line of enquiry culminated in the discovery of the 37kD protein initially called ‘lipocortin’ or ‘macrocortin’, which was later renamed AnxA1. AnxA1 inhibited the release of prostanoids apparently by inhibiting the action of PLA_2_ (at the time, cPLA2 had not been discovered) [79]. AnxA1 was successively purified to homogeneity from peritoneal lavage fluid obtained from rats treated with GCs [80] and subsequently cloned and sequenced [81]. AnxA1 is made up of an N-terminal domain that contains several putative phosphorylation sites at *serine*, *tyrosine*, and *threonine* residues essential for glycosylation, transglutamination, and proteolysis [82] and 346 amino acids [83]. AnxA1 is found in many tissues, including the lungs, bone marrow, and intestine, at concentrations of <50 ng/mL, with the highest levels reported to be in the seminal fluid (150 μg/mL). Although found in many differentiated cells and tissues, AnxA1 makes up 2–4% of the total cytosolic protein in some cells, such as PMN [84].

The precise underlying mechanism of GCs that govern AnxA1 gene transcription is yet to be fully elucidated, but it is interesting to note that although the AnxA1 promoter does not possess a recognised sequence for GC effects, it does bind to IL-6, thereafter inducing the expression of AnxA1. Subsequently, translocation of AnxA1 to the cell membrane and its secretion extracellularly, regulates the binding and attachment of leukocytes onto the endothelial cell surface [85].

An intricate association between AnxA1 and GCs was uncovered when it was shown that AnxA1 modulates the GC-induced secretion of the adrenocorticotrophic hormone (ACTH) from the anterior pituitary gland [86,87]. Congruently, it was also demonstrated that AnxA1 levels in murine peripheral blood leukocytes were raised two- to threefold within 2 h of steroid treatment, and this effect was blocked by the GC receptor antagonist RU486 [84].

GCs induce rapid nongenomic mobilisation and secretion of AnxA1 at the cell surface and a slower (2–4 h) upregulation of the AnxA1 gene transcription through genomic mechanisms during the inflammatory process [88]. The inhibition of eicosanoids by GCs could be classified based on AnxA1 dependency: first, rapid exposure of GCs via a nongenomic mechanism prevents the activation of cPLA_2_ through an AnxA1-dependent mechanism, and second, a more delayed exposure of GCs downregulates Cox-2 mRNA through an AnxA1-independent mechanism [89].

Exogenous recombinant AnxA1 peptides mimic the corticosteroid suppression of monocyte functions, such as superoxide generation [90,91] and autoimmune T lymphocyte proliferation [92]. Studies have also shown that AnxA1 plays an important role in mediating GCs’ antipyretic actions in rabbits [93] and hyperalgesia regulated by PG release in a murine model [94].

An important observation also shows that leukocytes obtained from patients with Addison’s disease, which is associated with reduced cortisol levels, have decreased intracellular levels of AnxA1, whilst patients with Cushing’s syndrome, which have with higher cortisol levels, have an increased expression of AnxA1, further corroborating the relationship between AnxA1 and GCs in disease [95]. Hence, during the inflammatory process, GCs positively induce the secretion of AnxA1, further activating innate immune cells, whilst limiting the proinflammatory response.

A substantial body of evidence shows that, indeed, AnxA1 is a bona fide mediator of the anti-inflammatory actions of GCs. Antisense and immunoneutralisation of the AnxA1 system blocks or inhibits the acute effect of GCs in several systems. Studies performed on AnxA1^−/−^ mice showed that they were more susceptible to inflammatory stimuli, COX-2 mRNA, and protein levels were constitutively increased, and more importantly, GCs were largely ineffective in these models [96], reiterating the homeostatic ability of AnxA1 to resolve inflammation. Interestingly, a study by Rigas and colleagues demonstrated that AnxA1 mediates the ability of a derivative of aspirin, nitric-oxide-donating aspirin (NO-ASA), to suppress NF-κB activation [38,97]. Whilst AnxA1 mediates some of the effects of potent GCs and some modified NSAIDs, it is uncertain whether its inhibitory action on NF-κB activation is the main (or only) mediator of their pharmacological effects.

AnxA1 acts extracellularly. The first step in AnxA1 release is phosphorylation by the PKC of *serine^27^*, which promotes membrane localisation and secretion. Once externalised, AnxA1 acts on extracellular receptors to produce its anti-inflammatory effects. These have been characterised as being members of the FPR receptor family. Studies with (the murine equivalent of) FPR-2 knockout mice have demonstrated once again an increased susceptibility to inflammation and a diminution of GC effects. Studies have established a strong association between the anti-inflammatory potency of GCs and the induction of AnxA1 and NF-κB inhibition, thus tempting the speculation that the ability of a compound to induce AnxA1 may determine its anti-inflammatory potency. If this is proven to be the case, one may predict that defects in AnxA1 synthesis or action may be responsible for the progression of inflammation.

Peptidomimics of AnxA1, such as N-acetyl 2-26 N-terminal peptide, have GC/AnxA1-like actions in a number of models, although they do not have quite the same specificity for the FPR isoforms that AnxA1 does, as the response of the receptors is ligand biased. Full-length AnxA1 was observed to activate the ALX/FPR2 homodimerisation, but its derivative Ac2-26 was shown to activate all members of the FPR family [98] and induce FPR1-ALX/FPR2 heterodimerisation [99]. The promiscuous ALX/FPR2 is bound by selective agonists that induce AnxA1 phosphorylation and mobilisation, leading to an anti-inflammatory and proresolving cascade [100].

A particularly interesting example of the importance of the AnxA1 system has come to light during studies with mast cells, which inadvertently explains the effect of another hitherto mysterious group of antiallergic drugs, the cromones or mast cell stabilisers [101]. These drugs have been shown to exert their antiallergic and anti-inflammatory activities through the secretion of the AnxA1 protein [102,103]. Sinniah et al. showed that bone-marrow-derived mast cells (BMDMCs) isolated from AnxA1^−/−^ mice were insensitive to the inhibitory effects of cromones, but interestingly, these cells retained their sensitivity to the inhibitory action of human recombinant AnxA1 (hu-r-ANX-A1) (Figure 1) [104]. Mast cells are abundant sources of AnxA1, and treatment with anti-inflammatory GCs, such as dexamethasone, or activation of the cell provokes a rapid induction of AnxA1 mRNA and/or a new protein. The ‘rapid’ GC response in mast cells and the antiallergic effects of the cromones were due to the release of preformed AnxA1 [104]. Yazid et al. showed that these drugs can act synergistically to inhibit the PP2A phosphatase enzyme that normally limits the activation of PKC, thereby prolonging the time course of activation of AnxA1 and increasing its release and hence its extracellular inhibitory action at FPR receptors (Figure 2). This mechanism underlies the ability of these drugs to inhibit histamine release from IgE-challenged human mast cells [105] and neutrophil migration and to mitigate intestinal reperfusion damage in the mouse [106] as well as to inhibit eicosanoid production [107].

Investigations into the roles and mechanisms of AnxA1 in various disease models in inflammation and cancer would further clarify the protein’s involvement in pathophysiology. Studies have shown that increased expression of AnxA1 could enhance the carotid atherosclerotic plaque stability, hence preventing plaque complications and progression [111]. The severity of liver fibrosis is also correlated with the elevation of the AnxA1 level, indicating that anti-inflammatory and proresolution effects are initiated. Sena et al. showed that AnxA1 expression is higher in macrophages isolated from patients with inflammatory bowel disease (IBD) during disease resolution [112,113]. In addition, AnxA1 was also found to be elevated in neutrophils and monocytes after skin injury, suggesting a role in wound formation at the beginning of trauma [114]. Neurological conditions are often characterised by the damaged blood–brain barrier (BBB); for example, it is interesting to note that AnxA1 has been demonstrated to reverse BBB damage triggered by inflammation or metabolic dysregulation and, hence, is able restore the normal function of BBB [115]. Conversely, AnxA1 expression in cancer is tissue specific [116]. Studies have shown that AnxA1 is highly expressed in pancreatic cancer, colorectal cancer, glioma, and liver cancer; however, it is downregulated in thyroid cancer, prostate cancer, and cervical cancer [117].

## 3. Conclusions

During the past half century, we have made impressive strides in our efforts to understand and to treat one of the commonest human afflictions, inflammatory disease. However, it is a never-ending quest: an understanding of how the body initiates and resolves inflammation using a battery of lipid and protein mediators may prove key to new therapeutic developments in the future.

## Figures and Tables

**Figure 1 cells-10-03524-f001:**
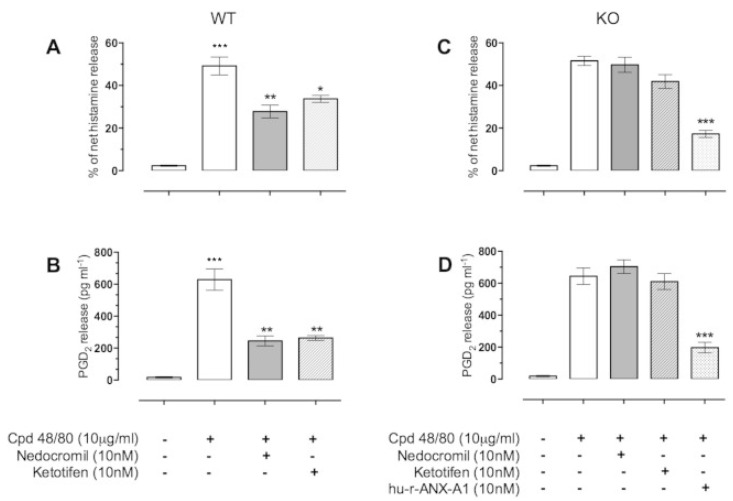
The inhibitory action of the mast cell ‘stabilisers’ nedocromil and ketotifen is dependent upon the presence of the mast cell AnxA1. BMDMCs from both AnxA1^+/+^ (WT) (panels **A**,**B**) and AnxA1^−/−^ (KO) (panels **C**,**D**) mice were cultured and prepared as described. Cells were stimulated with compound (Cpd) 48/80 (10 μg/mL) prior to treatment with either nedocromil (10 nM) or ketotifen (10 nM) and, in the case of the AnxA1^−/−^ cells, 10 nM hu-r-AnxA1. Data are expressed as mean ± SEM of *n* = 3 experiment (*** *p* < 0.001, ** *p* < 0.01, * *p* < 0.05) (figure reproduced with permission from the rights holder, Elsevier) [104].

**Figure 2 cells-10-03524-f002:**
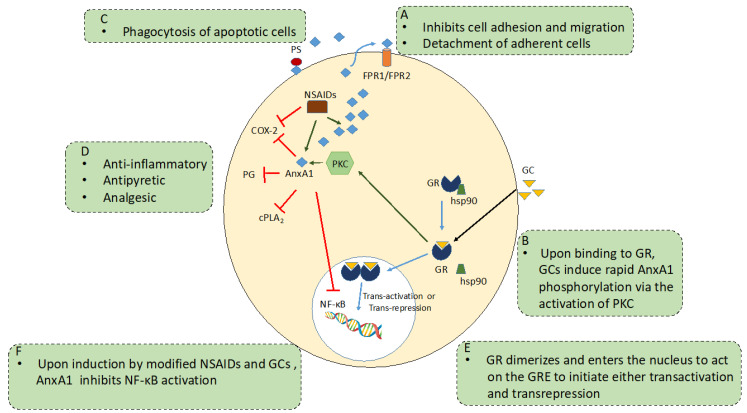
The principal mechanisms of AnxA1 anti-inflammatory actions. (**A**) Exogenous AnxA1 binds to formyl peptide receptors (FPR) to inhibit cell adhesion, migration and induce detachment of adherent cells [108]. (**B**) AnxA1 expression and release is up-regulated with GC treatment through the GC receptor (GR) either through genomic or non-genomic mechanisms, which contributes to the anti-inflammatory effects of AnxA1. GCs induce rapid AnxA1 phosphorylation via the activation of PKC and initiate the membrane translocation of AnxA1 molecule [72]. (**C**) AnxA1 is recruited to the cell surface, where it binds to phosphatidylserine (PS) and mediates the phagocytosis of apoptotic bodies [109]. (**D**) AnxA1 inhibited the cytosolic phospholipase A_2_ (cPLA_2_), prostaglandins and cyclooxygenase 2 (COX-2), thus exhibiting anti-inflammatory, anti-pyretic and anti-analgesic activities. (**E**) GR dimerize and translocate to the nucleus and binds to the GC Response Element (GRE) to initiate trans-activation or trans-repression [110]. (**F**) AnxA1 induced by GC and modified NSAIDs, binds to NF-κB to inhibit its activation [97].

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
