# Peer review of "From NSAIDs to Glucocorticoids and Beyond"

_cells, 2021, doi:10.3390/cells10123524_

Round 1

Reviewer 1 Report

Sinniah and cols. proposed to review NSAIDs and glucocorticoids. A very interesting/attractive historic approach was delineated mainly focusing on pain, since Hippocrates until now and reinforcing new uses of old drugs with examples. In the manuscript, the authors merged mechanisms of action and current discoveries and make the document interesting to be read. The revision was very well delineated and rheumatoid arthritis was undertaken as a model of inflammatory disease.

The article is very well-organized, and the literature was good synthesizing. Considering the approach taken by the authors, the review is complete and there is no doubt about the current relevance of anti-inflammatory drugs. The references are appropriate, and no gap was identified in the manuscript.

I was wondering and I only have a suggestion. To reinforce the importance of discoveries concerning the mechanism of actions of currently used molecules, such as glucocorticoids, may be interesting to include a paragraph considering the potential anti-inflammatory and pro-resolutive actions of AnxA1 in other inflammatory conditions, such as neurologic conditions, gut diseases, sepsis, vascular diseases. I think that it could maybe include as future perspectives.

And, finally, I congratulate you on this wonderful review which I believe will contribute to the understanding of anti-inflammatory drugs' evolution.

Author Response

Sinniah and cols. proposed to review NSAIDs and glucocorticoids. A very interesting/attractive historic approach was delineated mainly focusing on pain, since Hippocrates until now and reinforcing new uses of old drugs with examples. In the manuscript, the authors merged mechanisms of action and current discoveries and make the document interesting to be read. The revision was very well delineated and rheumatoid arthritis was undertaken as a model of inflammatory disease.

The article is very well-organized, and the literature was good synthesizing. Considering the approach taken by the authors, the review is complete and there is no doubt about the current relevance of anti-inflammatory drugs. The references are appropriate, and no gap was identified in the manuscript.

I was wondering and I only have a suggestion. To reinforce the importance of discoveries concerning the mechanism of actions of currently used molecules, such as glucocorticoids, may be interesting to include a paragraph considering the potential anti-inflammatory and pro-resolutive actions of AnxA1 in other inflammatory conditions, such as neurologic conditions, gut diseases, sepsis, vascular diseases. I think that it could maybe include as future perspectives.

Thank you for the positive feedback. We really appreciate your suggestions. We have added a paragraph to incorporate the anti-inflammatory and pro-resolution of Anx-A1 in the other inflammatory conditions as well. Please refer to line 406-422.

And, finally, I congratulate you on this wonderful review which I believe will contribute to the understanding of anti-inflammatory drugs' evolution.

Thank you so much for the kind comments and suggestion.

Reviewer 2 Report

The subject of this manuscript is very important and the authors have a respected history and expertise in anti-inflammatory drugs. However, the manuscript is not very exciting and does not have much updated and new description regarding NSAIDs and glucocorticoids drugs. I would suggest that the authors complement this manuscript with a more in-depth description of the mechanism of action of both groups, including the immunosuppressants that are referred to as "beyond" in this review. Furthermore, it could add more updated applications of these anti-inflammatories, such as the new  application of dexamethasone in COVID-19  treatment, or application of aspirin and other NSAIDs in neurological disturbances such as Alzheimer's disease, among others.

In addition, I suggest making revisions and corrections of some concepts describe in this manuscript:

- In line 38, referring to the formation of gaps between venules and capillaries and passage of fluid and cells. Increased cell permeability refers to edema caused by fluid leakage into the inflammatory focus. Cellular influx, on the other hand, is due to the process of marginalization, fixation, rolling of leukocytes together with the endothelial cells that border the lesion site. Cells migrate through the interaction of specifically expressed adhesion molecules and migrate per se to the interstitium;

-In lines 110 and 111, the sequential history of the development and synthesis of acetylsalicylic acid, or aspirin, should be better described. What is the meaning of "the first mee-too" in this context?

-The paragraph covering lines 174 to 185 has no references;

-NSAID has been pointed out as a possible adjunct treatment for tumor processes, as the literature has shown. Thus, I thought that the authors' doubts about this subject (lines 162 to 164) should be re-elaborated. Mechanisms are proposed, as studies that have showed that aspirin prevents colorectal cancer through inhibition of NF-κB and signaling pathways. New targets of Aspirin and Salicylic Acid have been proposed, such as Cyclin A2 and CDK2, among others, as a preventive action mechanism against cancer On the other hand, aspirin can also acetylate the inflammatory COX-2 isoform, leading to the production of pro-resolving lipid mediators, derived from omega-3;

-Line 261 correct "monomeric GR" to "homodimeric GR".

Author Response

The subject of this manuscript is very important and the authors have a respected history and expertise in anti-inflammatory drugs. However, the manuscript is not very exciting and does not have much updated and new description regarding NSAIDs and glucocorticoids drugs. I would suggest that the authors complement this manuscript with a more in-depth description of the mechanism of action of both groups, including the immunosuppressants that are referred to as "beyond" in this review. Furthermore, it could add more updated applications of these anti-inflammatories, such as the new  application of dexamethasone in COVID-19  treatment, or application of aspirin and other NSAIDs in neurological disturbances such as Alzheimer's disease, among others.

Thank you for the suggestions. Although immunosuppressants would provide a greater insight to the manuscript, we do feel that this is out of the scope of our review paper.

We have added the new application dexamethasone in COVID-19 treatment (please refer to line 256-262)and also have explained the association of aspirin with Alzheimer’s disease (please refer to line 180-183).

In addition, I suggest making revisions and corrections of some concepts describe in this manuscript:

- In line 38, referring to the formation of gaps between venules and capillaries and passage of fluid and cells. Increased cell permeability refers to edema caused by fluid leakage into the inflammatory focus. Cellular influx, on the other hand, is due to the process of marginalization, fixation, rolling of leukocytes together with the endothelial cells that border the lesion site. Cells migrate through the interaction of specifically expressed adhesion molecules and migrate per se to the interstitium;

Thanks for the suggestion, we have incorporated your comment in our manuscript. Please kindly refer to line 39-48.

-In lines 110 and 111, the sequential history of the development and synthesis of acetylsalicylic acid, or aspirin, should be better described. What is the meaning of "the first mee-too" in this context?

The sequential history of the development and synthesis of aspirin was described. Please refer to line 118-124.The term "me-too drug" refers to a medication that is similar to a pre-existing drug, usually by making minor modifications to the prototype, reflected in slight changes in the profiles of side effects or activity, and used to treat conditions for which drugs already exist. (https://en.wikipedia.org/wiki/Me-too_drug)

-The paragraph covering lines 174 to 185 has no references;

Thank you for the observation. The relevant references have been added in the manuscript. Please refer to the line 193-204.

-NSAID has been pointed out as a possible adjunct treatment for tumor processes, as the literature has shown. Thus, I thought that the authors' doubts about this subject (lines 162 to 164) should be re-elaborated. Mechanisms are proposed, as studies that have showed that aspirin prevents colorectal cancer through inhibition of NF-κB and signalling pathways. New targets of Aspirin and Salicylic Acid have been proposed, such as Cyclin A2 and CDK2, among others, as a preventive action mechanism against cancer On the other hand, aspirin can also acetylate the inflammatory COX-2 isoform, leading to the production of pro-resolving lipid mediators, derived from omega-3;

Thanks for the comment. We have rephrased the paragraph and have incorporated your suggestions. Please refer to line 175-183.

-Line 261 correct "monomeric GR" to "homodimeric GR".

The correction has been made. Please refer to line 287.

Reviewer 3 Report

Sininiah and colleagues provide an interesting and historical overview of concepts and factors associated with the control of inflammation. In the later part of this manuscript, the authors provided an update on the mechanism of action of annexin A1, an important glucocorticoid-induced molecule with pro-resolving effects. Overall, the manuscript is well written and contains relevant information that is of interest to the readers in the field of inflammation and inflammation resolution. 

Minor comments:

  • Based on the historical overview covered in this review, it would be interesting to add a timeline citing the main achievements in the field of inflammation, anti-inflammation, and inflammation resolution.
  • Page 2, lines 52-54. It is written: “… endogenous lipid molecules that mediate resolution of inflammation, by inhibiting neutrophil recruitment and extravasation, attenuating phagocytosis of apoptotic neutrophils as a mechanism of the clearance of inflammation and tissue repair”. It seems to be the opposite, that these lipid molecules increase the phagocytosis of apoptotic neutrophils which contribute to the resolution of inflammation.
  • Page 6, lines 299-302. The relationship between IL-6, glucocorticoids and annexin-A1 is unclear. Can GC bind to the promoter region of IL-6 for the regulation of Anx-A1 expression?

Author Response

Sininiah and colleagues provide an interesting and historical overview of concepts and factors associated with the control of inflammation. In the later part of this manuscript, the authors provided an update on the mechanism of action of annexin A1, an important glucocorticoid-induced molecule with pro-resolving effects. Overall, the manuscript is well written and contains relevant information that is of interest to the readers in the field of inflammation and inflammation resolution. 

Minor comments:

  • Based on the historical overview covered in this review, it would be interesting to add a timeline citing the main achievements in the field of inflammation, anti-inflammation, and inflammation resolution.

Thank you so much for your kind suggestion. We have decided not to add an extra table of timeline curating the main achievements in the field of inflammation, anti-inflammation and resolution, because several other papers (Vane JR. The history of anti-inflammatory drugs and their mechanism of action. 1996. New targets in Inflammation., and Serhan CN. The resolution of inflammation: the devil in the flask and in the details. 2011. FASEB) have outlined the timeline and hence, it might be redundant to report similar findings.

  • Page 2, lines 52-54. It is written: “… endogenous lipid molecules that mediate resolution of inflammation, by inhibiting neutrophil recruitment and extravasation, attenuating phagocytosis of apoptotic neutrophils as a mechanism of the clearance of inflammation and tissue repair”. It seems to be the opposite, that these lipid molecules increase the phagocytosis of apoptotic neutrophils which contribute to the resolution of inflammation.

Thanks for your observation. We have amended accordingly. Please refer to line 59.

  • Page 6, lines 299-302. The relationship between IL-6, glucocorticoids and annexin-A1 is unclear. Can GC bind to the promoter region of IL-6 for the regulation of Anx-A1 expression?

The paragraph has been rephrased for better understanding (please refer to the line 326-330). Thank you.

Round 2

Reviewer 2 Report

I agree with the corrections and new text insertions made to this manuscript by the authors.

The only indication of correction would be adequacy of the reference described in line 428.